# STRUCTURED PRUNING MEETS ORTHOGONALITY

## ABSTRACT

Several recent works empirically found finetuning learning rate is crucial to the final performance in structured neural network pruning. It is shown that the *dynamical isometry* broken by pruning answers for this phenomenon. How to develop a filter pruning method that maintains or recovers dynamical isometry *and* is scalable to modern deep networks remains elusive up to now. In this paper, we present *orthogonality preserving pruning* (OPP), a regularization-based structured pruning method that maintains the dynamical isometry during pruning. Specifically, OPP regularizes the gram matrix of convolutional kernels to encourage kernel orthogonality among the important filters meanwhile driving the unimportant weights towards zero. We also propose to regularize batch-normalization parameters for better preserving dynamical isometry for the whole network. Empirically, OPP can compete with the *ideal* dynamical isometry recovery method on linear networks. On non-linear networks (ResNet56/VGG19, CIFAR datasets), it outperforms the available solutions *by a large margin*. Moreover, OPP can also work effectively with modern deep networks (ResNets) on ImageNet, delivering encouraging performance in comparison to many recent filter pruning methods. To our best knowledge, this is the *first* method that effectively maintains dynamical isometry during pruning for *large-scale* deep neural networks.

## 1 INTRODUCTION

Neural network pruning aims to remove the parameters without seriously compromising the performance. It normally consists of three steps (Reed, 1993; Han et al., 2015; 2016b): pretrain a dense model; prune the unnecessary connections or neurons with some criteria; finetune to regain performance. Pruning is usually categorized into two groups, unstructured pruning (a.k.a. element-wise pruning) and structured pruning (a.k.a. filter pruning or channel pruning). The former chooses a scalar weight as the basic pruning element; the latter chooses a 3d filter as the basic pruning element. In general, structured pruning is more favored for acceleration on commodity hardware because of its consequent regular sparsity; unstructured pruning results in irregular sparsity, which can be considerable without performance degradation but hard to exploit for acceleration if not using customized hardware (Han et al., 2016a; 2017).

Recent structured pruning works (Renda et al., 2020; Le & Hua, 2021; Wang et al., 2021a) showed an interesting phenomenon: in the finetuning step, using a larger learning rate (LR) helps to achieve a significantly better final performance (*e.g.*, ResNet34 pruned at speedup $1.32\times$ can be improved by over 1% top-1 accuracy on ImageNet (Deng et al., 2009) using finetuning LR `1e-2` vs. `1e-3`). The reason behind is shown by (Wang et al., 2021a) relevant to *dynamical isometry* (Saxe et al., 2014), a nice property of neural networks that are easy to train without gradient vanishing or explosion (Glorot & Bengio, 2010; Pascanu et al., 2013). They mainly made two observations for explanation. First, the weights removal operation immediately breaks the dynamical isometry of the pretrained network. Second, SGD training in finetuning can help recover it; a larger LR help recover it faster and better, thus making the final performance stronger. Although (Wang et al., 2021a) provided a sound explanation, a more practical issue is how to recover the broken dynamical isometry or maintain it during pruning. In this regard, (Wang et al., 2021a) proposed to apply weight orthogonalization based on QR decomposition (Trefethen & Bau III, 1997; Mezzadri, 2006) to the pruned model. However, the method was shown to only work for *linear* networks. On modern deep convolutional neural networks (CNNs), the method is still far from satisfactory.

In this paper, we present *orthogonality preserving pruning* (OPP), a new filter pruning method that maintains dynamical isometry well during pruning. The main idea of OPP is to promote kernel orthogonality among the kept filters meanwhile pushing the weights to be pruned rather close to zero. By doing so, the subsequent weights removal operation will barely hurt the dynamical isometry of the network. Specifically, we propose to regularize the gram matrix of weights: all the entries representing the correlation between the pruned filters and the others are encourage to diminish to zero. This is the first technical contribution of our method. The second one lies in how to treat the cross-correlation entries of kept filters. Inspired by the proposed orthogonalization initialization in (Saxe et al., 2014), we add a kernel orthogonality term to the kept filters, which promotes dynamical isometry during pruning.

In addition, modern deep models are typically equipped with batch normalization (BN) (Ioffe & Szegedy, 2015). However, previous filter pruning papers rarely explicitly take BN into account (except two (Liu et al., 2017; Ye et al., 2018); the differences between our work and theirs will be discussed in Sec. 3.2) to mitigate the side effect when it is removed because its associated filter is removed. By the idea of preserving dynamical isometry, they should not be ignored since their removal breaks dynamical isometry as well. Therefore, we propose to regularize the learnable parameters of BN to minimize the influence of its absence.

Empirically, the proposed pruning algorithm is easy to implement and delivers encouraging results compared to many existing filter pruning methods. Notably, this is the first method that effectively maintains dynamical isometry during pruning on modern *large-scale* deep networks.

**Contributions**. We make three contributions in this paper:

- We present the *first* filter pruning method (orthogonality preserving pruning) that can effectively maintain dynamical isometry when pruning modern deep networks, through a customized weight gram matrix as regularization target.
- Apart from weight regularization, we also propose to regularize the batch normalization parameters to better maintain dynamical isometry. This has been overlooked by most previous pruning papers, while we show it is an indispensable part if we aim to maintain dynamical isometry for the *whole* network.
- Practically, the proposed method is scalable to modern large-scale deep neural networks (*e.g.*, ResNets) and datasets (*e.g.*, ImageNet). It achieves promising pruning performance in comparison to many state-of-the-art filter pruning methods.

## 2 RELATED WORK

**Neural network pruning**. In terms of pruning granularity, pruning methods mainly fall into structured pruning (a.k.a. filter pruning or channel pruning) (Li et al., 2017; Wen et al., 2016; He et al., 2017; 2018a; Wang et al., 2021c) and unstructured pruning (a.k.a. element-wise pruning) (Han et al., 2015; 2016b; LeCun et al., 1990; Hassibi & Stork, 1993; Singh & Alistarh, 2020). Structured pruning results in *regular* sparsity after pruning, easy to be translated to acceleration on commodity hardware. In contrast, unstructured pruning produces *irregular* sparsity, beneficial to compression while hard to leverage for practical acceleration (Wen et al., 2016; Wang et al., 2019b) unless with special hardware support (Han et al., 2016a; 2017). For more comprehensive coverage, we recommend surveys (Sze et al., 2017; Cheng et al., 2018a;b; Deng et al., 2020; Wang et al., 2021b). We focus on *filter pruning* in this work for acceleration.

Most pruning papers focus on finding a better pruning criterion to select unimportant parameters to remove. The solutions primarily follow two paradigms (Reed, 1993): regularization-based and importance-based. The former selects unimportant parameters by adding a sparsity-inducing penalty term, which is jointly optimized with the original loss objective function (*e.g.*, (Wen et al., 2016; Lebedev & Lempitsky, 2016; Louizos et al., 2018; Liu et al., 2017; Ye et al., 2018)). The latter selects unimportant parameters through certain derived mathematical formula (*e.g.*, (LeCun et al., 1990; Hassibi & Stork, 1993; Han et al., 2015; 2016b; Li et al., 2017; Molchanov et al., 2017; 2019)). Notably, there is *no* strict boundary between the two paradigms. Several works (Ding et al., 2018; Wang et al., 2019b; 2021c) manage to get the best from both worlds – they select unimportant weights by an importance criterion *and* add a penalty term for sparsity as well. Our method in this paper also belongs to this group, while it achieves stronger performances.

**Dynamical isometry and orthogonality**. Dynamical isometry was first introduced by (Saxe et al., 2014), where it can be achieved (for linear MLP models) by the orthogonality of weight matrix at initialization. Recent works on this topic mainly focus on how to maintain dynamical isometry *during training* instead of only for initialization (Xie et al., 2017; Huang et al., 2018; Bansal et al., 2018; Huang et al., 2020; Wang et al., 2020). These methods are developed independent of pruning, thus not directly relevant to the proposed method in this work. However, the insights from these works inspire us to our proposed approach (see Sec. 3.2) and possibly more in the future.

**Pruning + dynamical isometry**. One particular paper that inspires us to this work is (Wang et al., 2021a), where the authors leverage dynamical isometry (Saxe et al., 2014) to explain the performance boosting effect (also noted by (Renda et al., 2020; Le & Hua, 2021)) of using a larger fine-tuning LR in pruning: *pruning hurts dynamical isometry; finetuning can recover dynamical isometry and a larger LR helps recover it faster (and possibly better), thus delivering superior final performance.* The explanation further clears the mystery about the value of network pruning (Liu et al., 2019; Crowley et al., 2018). Previously, (Liu et al., 2019; Crowley et al., 2018) found the small models can be trained from scratch with comparable accuracy to the counterparts pruned from a pretrained large model, thus they argue no value of inheriting weights in structured pruning. (Wang et al., 2021a) pointed out that this argument is actually built upon sub-optimal finetuning LR setups. With the proper ones, filter pruning consistently outperforms training from scratch.

## 3 METHODOLOGY

### 3.1 PRELIMINARIES: DYNAMICAL ISOMETRY AND ORTHOGONALITY

The definition of dynamical isometry is that the input-output Jacobian of a network has as many singular values (JSVs) as possible close to 1 (Saxe et al., 2014). With it, the error signal can preserve its norm under propagation without serious amplification or attenuation, which in turn helps the convergence of (very deep) networks. For a single fully-connected layer $W$, a sufficient and necessary condition to realize dynamical isometry is orthogonality, *i.e.*, $W^T W = I$, as shown below,

$$\mathbf{y} = W\mathbf{x},$$
$$||\mathbf{y}|| = \sqrt{\mathbf{y}^T \mathbf{y}} = \sqrt{\mathbf{x}^T W^T W \mathbf{x}} = ||\mathbf{x}||, \ \textit{iff. } W^T W = I, \tag{1}$$

where $I$ stands for identity matrix. Orthogonality of a weight matrix can be easily realized by matrix orthogonalization techniques such as QR decomposition (Trefethen & Bau III, 1997; Mezzadri, 2006). *Exact* (namely all the Jacobian singular values are exactly 1) dynamical isometry can be achieved for *linear* networks since multiple linear layers essentially reduce to a single 2d weight matrix. In contrast, the convolutional and non-linear cases are much complicated. Previous work (Wang et al., 2021a) has shown that merely considering convolution or ReLU (Nair & Hinton, 2010) renders the weight orthogonalization method much less effective in terms of recovering dynamical isometry after pruning, let alone considering modern deep networks with BN (Ioffe & Szegedy, 2015) and residuals (He et al., 2016). The primary goal of our paper is to bridge this gap.

Following the seminal work of (Saxe et al., 2014), several papers propose to maintain orthogonality *during training* instead of sorely for the initialization. There are primarily two groups of orthogonalization methods for convolutional neural networks: kernel orthogonality (Xie et al., 2017; Huang et al., 2018; 2020) and orthogonal convolution (Wang et al., 2020):

$$KK^T = I \Rightarrow \mathcal{L}_{orth} = KK^T - I, \triangleleft \textbf{ kernel orthogonality}$$
$$\mathcal{K}\mathcal{K}^T = I \Rightarrow \mathcal{L}_{orth} = \mathcal{K}\mathcal{K}^T - I, \triangleleft \textbf{ orthogonal convolution} \tag{2}$$

where clearly the difference lies in the weight matrix $K$ vs. $\mathcal{K}$. **(1)** $K$ denotes the original weight matrix in a convolutional layer. Weights of a conv layer make up a 4d tensor $\mathbb{R}^{N \times C \times H \times W}$, where $N$ stands for the number of output channels, $C$ for the number of input channels, $H$ and $W$ for the height and width of conv kernel. Then, $K$ is a reshaped version of the 4d tensor: $K \in \mathbb{R}^{N \times CHW}$ (if $N < CHW$; otherwise, $K \in \mathbb{R}^{CHW \times N}$). **(2)** In contrast, $\mathcal{K} \in \mathbb{R}^{NH_{fo}W_{fo} \times CH_{fi}W_{fi}}$ stands for the doubly block-Toeplitz (DBT) matrix representation of $K$ ($H_{fo}$ stands for the output feature map height, $H_{fi}$ for the input feature map height. $W_{fo}$ and $W_{fi}$ can be inferred the same way for width).

It was showed by (Wang et al., 2020) that orthogonal convolution is more effective than kernel orthogonality (Xie et al., 2017) in that the latter is only a necessary but insufficient condition of the

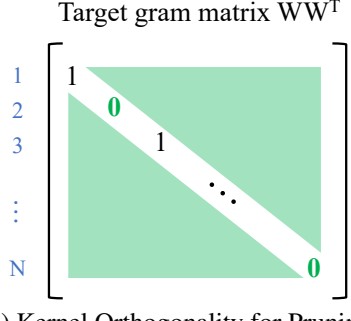

(a) Kernel Orthogonality  (b) Kernel Orthogonality for Pruning

Figure 1: Regularization target comparison between the proposed scheme (b) and kernel orthogonality (a). Green part stands for zero entries (this figure is best viewed in color). Index 1 to N denotes the filter indices. In (b, c), filter 2 and N are the unimportant filters to be removed. **(a)** Regularization target of pure kernel orthogonality (an identity matrix), no pruning considered. **(b)** Regularization target when considering both kernel orthogonality and filter pruning.

former. In this work, we will evaluate both methods to see how effective they are in maintaining/recovering the broken dynamical isometry.

## 3.2 ORTHOGONALITY PRESERVING PRUNING (OPP)

The proposed method is made up of two parts with the same goal: maintaining as much dynamical isometry during pruning as possible. First, we explain how to employ regularization to do so for convolutional (Conv) or fully-connected (FC) layers. Second, we propose to regularize batch normalization layers given their prevailing use as a standard component in deep networks nowadays.

**(1) Orthogonality meets filter pruning**. From previous works (Lee et al., 2020; Wang et al., 2021a), we know recovering the broken dynamical isometry incurred by pruning is imperative. Considering orthogonality regularization can encourage dynamical isometry, a pretty straightforward solution is to build upon the existing kernel orthogonality regularization schemes. Specifically, kernel orthogonality regularizes the weight gram matrix to be close to identity matrix (see Fig. 1(a)). In our case, we aim to remove some filters, so naturally we can regularize the weight gram matrix to be close to a *partial identity matrix*, with the diagonal entries at the pruned filters zeroed (see Fig. 1(b)).

Specifically, for the $l$-th layer, we sort the filters by their $L_1$-norms and select those with the least $L_1$-norms as unimportant filters (denoted as set $S_l$). Then, the proposed regularization term is,

$$\mathcal{L}_1 = \sum_{l=1}^{L} ||W_l W_l^T \odot -\hat{I}||_F^2, \ \hat{I}_{jj} = 0 \text{ if } j \in S_l, \text{ else } 1, \tag{3}$$

where $W$ denotes the weight matrix; $\hat{I}$ represents the partial identity matrix, whose diagonal entries of pruned filters are 0 instead of 1; and $||\cdot||_F$ denotes the Frobenius norm.

**(2) BN regularization**. There are mainly two kinds of learnable layers in a deep neural network: Conv/FC and BN. The above regularization (Eq. (3)) maintains dynamical isometry for Conv/FC layers. By the idea of preserving dynamical isometry for the *whole* network, BN is not ignorable since removing filters will change the internal feature distributions. If the learned BN statistics do not change accordingly, the error will accumulate. Dynamical isometry will be impaired still. Look at the following BN formulation (Ioffe & Szegedy, 2015),

$$f = \gamma \frac{W * X - \mu}{\sqrt{\sigma^2 + \epsilon}} + \beta, \tag{4}$$

where $*$ stands for convolution; $\mu$ stands for the running mean, $\sigma^2$ for the running variance; $\epsilon$ is a small amount for numerical stability; the two learnable parameters are $\gamma$ and $\beta$. When a filter is zeroed out, the output feature map channel is *not* zeroed out. These non-zero values are fed into the subsequent layers and play a part in the network. The dynamical isometry of other parameters will depend on these values. Once these values are zeroed, dynamical isometry will be damaged,

---

**Algorithm 1** Orthogonality Preserving Pruning (OPP)

1: **Input**: Pretrained model $\Theta$, layer-wise pruning ratio $r_l$ of $l$-th layer, for $l \in \{1, 2, \cdots, L\}$.
2: **Input**: Regularization ceiling $\tau$, penalty coefficient update interval $K_u$, penalty granularity $\Delta$.
3: **Init**: Iteration $i = 0$. $\lambda_j = 0$ for all filter $j$. Set pruned filter indices $S_l$ by $L_1$-norm sorting.
4: **while** $\lambda_j \leq \tau$, for $j \in S_l$ **do**
5:     **if** $i \% K_u = 0$ **then**
6:         $\lambda_j = \lambda_j + \Delta$ for $j \in S_l$.           $\triangleleft$ Update regularization co-efficient in Eq. (6)
7:     **end if**
8:     Network forward, loss (Eq. (6)) backward, parameter update by stochastic gradient descent.
9:     Update iteration: $i = i + 1$.
10: **end while**
11: Remove filters in $S_l$ to obtain a smaller model $\Theta'$.
12: Finetune $\Theta'$ to regain accuracy.
13: **Output**: Finetuned model $\Theta'$.

---

causing (irrecoverable) performance degradation. To overcome this, we propose to regularize the $\gamma$ and $\beta$ of *pruned* feature map channels to zero. The penalty term can be formulated as

$$\mathcal{L}_2 = \sum_{l=1}^{L} \sum_{j \in S_l} \gamma_j^2 + \beta_j^2. \tag{5}$$

This regularization term is simple to implement in any deep learning frameworks, also ready to work with other filter pruning methods. The merits of this BN regularization will be demonstrated in our experiments (see Tab. 3). To sum, with the proposed regularization terms, the total error function is

$$\mathcal{E} = \mathcal{L}_{cls} + \frac{\lambda}{2}(\mathcal{L}_1 + \mathcal{L}_2), \tag{6}$$

where $\mathcal{L}_{cls}$ stands for the original classification loss. For the unimportant (or pruned) filters, the coefficient $\lambda$ *grows* gradually (by a predefined constant $\Delta$ per $K_u$ iterations, up to a ceiling $\tau$) during training to ensure the pruned parameters are rather close to zero (inspired by (Wang et al., 2019b; 2021c)). For the important (or remaining) filters, $\lambda$ is set to a constant ($10^{-3}$ in our experiments). Our algorithm can be summarized in Algorithm 1.

**Discussion**. Prior works (Liu et al., 2017; Ye et al., 2018) also propose to regularize BN for pruning. Our BN regularization method is *starkly different* from theirs. (1) In terms of motivation or goal, (Liu et al., 2017; Ye et al., 2018) regularize $\gamma$ to *learn* unimportant filters, namely, regularizing BN is to indirectly decide which filters are unimportant. In contrast, in our method, unimportant filters are decided by their $L_1$-norms. We adopt BN regularization for a totally different consideration – to mitigate the side effect of breaking dynamical isometry, which is not mentioned at all in their works. (2) In terms of specific technique, (Liu et al., 2017; Ye et al., 2018) only regularize the scale factor $\gamma$ (because it is enough to decide which filters are unimportant) while we regularize *both* the learnable parameters because only regularizing one still impairs dynamical isometry of the network. Besides, we employ different regularization strength for different parameters (by the group of important filters vs. unimportant filters), while (Liu et al., 2017; Ye et al., 2018) simply use the same penalty strength for *all* parameters – this is another key difference because regularizing all parameters (including those that are meant to be kept) will damage dynamical isometry, which is exactly what we want to avoid. In short, in either general motivation or specific technical details, our proposed BN regularization is *distinct* from previous works (Liu et al., 2017; Ye et al., 2018).

## 4 EXPERIMENTAL RESULTS

**Datasets and networks**. We first conduct analyses with MLP-7-Linear network on MNIST (Le-Cun et al., 1998). Then compare our method to other plausible solutions with ResNet56 (He et al., 2016)/VGG19 (Simonyan & Zisserman, 2015) on CIFAR10/100 (Krizhevsky, 2009). Next we evaluate our algorithm on the large-scale ImageNet dataset (Deng et al., 2009) with ResNet34/50 (He et al., 2016) and MobileNetV2 (Sandler et al., 2018). Finally, we conduct ablation study to show the efficacy of the proposed BN regularization. On ImageNet, we take the official PyTorch (Paszke et al.,

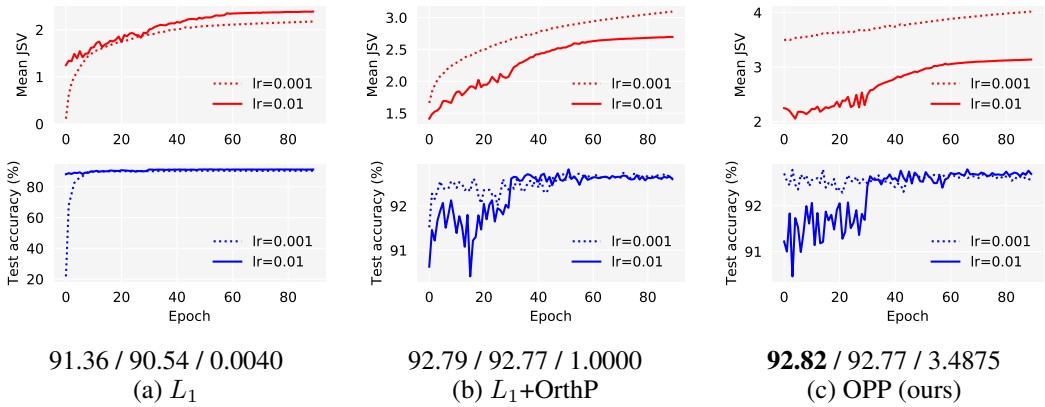

91.36 / 90.54 / 0.0040
(a) $L_1$

92.79 / 92.77 / 1.0000
(b) $L_1$+OrthP

**92.82** / 92.77 / 3.4875
(c) OPP (ours)

Figure 2: Mean JSV and test accuracy during finetuning with different setups (network: MLP-7-Linear, dataset: MNIST). $L_1$ refers to the $L_1$-norm pruning (Li et al., 2017). OrthP is the dynamical isometry recovery method proposed by (Wang et al., 2021a). Below each plot are, in order, the best accuracy of LR 1e-2, the best accuracy of LR 1e-3, and the mean JSV right after pruning (*i.e.*, without finetuning). LR 1e-2 and 1e-3 are short for two finetuning LR schedules following (Wang et al., 2021a): {0:1e-2, 30:1e-3, 60:1e-4, #epochs:90}, {0:1e-3, 45:1e-4, #epochs:90}. The accuracies are averaged by 5 random runs. For reference, the unpruned model has mean JSV 2.4987, test accuracy 92.77.

2019) pre-trained models[1] as base models to maintain comparability with other methods. On other datasets, we train our own base models with comparable or better accuracies than those reported in the original papers. *Our code and trained models will be made publicly available.*

**Training settings**. Given limited space, we defer the detailed hyper-parameter settings (*e.g.*, batch size, weight decay) to the Appendix. Important ones (*e.g.*, LR) will be mentioned in this main text.

**Comparison methods**. We compare with (Wang et al., 2021a), which is the only work proposing a method (called OrthP) to recover broken dynamical isometry for pruning *pretrained* models, as far as we know. Another method worth comparison is Approximated Isometry (AI) (Lee et al., 2020), which was initially proposed for recovering broken dynamical isometry on *randomly* initialized networks, but still, it can be used in case of pruning a pretrained model. Thus, we also compare with it. Besides, since maintaining orthogonality is the key and there are many existing orthogonality regularization papers (Xie et al., 2017; Wang et al., 2020; Huang et al., 2018; 2020; Wang et al., 2020), a straightforward solution is to combine them with $L_1$-norm pruning (Li et al., 2017) to see if they can help maintain or recover the broken dynamical isometry. There are two possible combination schemes: 1) apply orthogonal regularization methods *before* applying $L_1$-norm pruning, 2) apply orthogonal regularization methods *after* $L_1$-norm pruning, namely, in finetuning. Two representative orthogonality regularization methods are selected because of their proved effectiveness: kernel orthogonality (Xie et al., 2017) and convolutional orthogonality (Wang et al., 2020), so in total there are 4 combinations: $L_1$ + KernOrth (Xie et al., 2017), $L_1$ + OrthConv (Wang et al., 2020), KernOrth (Xie et al., 2017) + $L_1$, OrthConv (Wang et al., 2020) + $L_1$.

**Layer-wise pruning ratios**. We adopt manually *preset* pruning ratios in our method following (Li et al., 2017; He et al., 2018a; Wang et al., 2019b; 2021c) to maintain comparability with prior works. Moreover, presetting pruning ratios help us control irrelevant factors precisely, excluding their interference to our analyses. Please refer to the Appendix for a summary of all pruning ratios adopted in this work and the discussion about the reason of employing *fixed* pruning ratios.

**Comparison metrics**. (1) We compare the final test accuracy *after finetuning* with the similar FLOPs budget – this is currently the most prevailing metric to compare different filter pruning methods in classification. Concretely, we compare two settings: a relatively large finetuning LR (1e-2) and a small one (1e-3). We introduce these settings because previous works (Renda et al., 2020; Le & Hua, 2021; Wang et al., 2021a) showed that finetuning LR has a great impact on the final performance. From this metric, we can see how sensitive different methods are to the finetuning LR.

---

[1]https://pytorch.org/docs/stable/torchvision/models.html

Table 1: Test accuracy (%) comparison among different dynamical isometry maintenance or recovery methods on CIFAR10/100. "Scratch" stands for training from scratch. Each setting is randomly run for 3 times, mean (std) accuracies reported. "KernOrth" means Kernel Orthogonalization (Xie et al., 2017); "OrthConv" means Convolutional Orthogonalization (Wang et al., 2020). Two finetuning LR schedules are evaluated here: initial LR `1e-2` vs. `1e-3`. "Acc. diff." refers to the accuracy gap of LR `1e-2` over LR `1e-3`.

| **ResNet56 + CIFAR10**: Baseline accuracy 93.78%, Params: 0.85M, FLOPs: 0.25G | | | | | |
|---|---|---|---|---|---|
| Pruning ratio $r$ | 0.3 | 0.5 | 0.7 | 0.9 | 0.95 |
| Sparsity/Speedup | 31.14%/1.45× | 49.82%/1.99× | 70.57%/3.59× | 90.39%/11.41× | 95.19%/19.31× |
| | Initial | finetuning | LR 1e-2 | | |
| Scratch | 93.16 (0.16) | 92.78 (0.23) | 92.11 (0.12) | 88.36 (0.20) | 84.60 (0.14) |
| $L_1$ (Li et al., 2017) | 93.79 (0.06) | **93.51** (0.07) | 92.26 (0.17) | 88.71 (0.15) | 84.63 (0.28) |
| $L_1$ + OrthP (Wang et al., 2021a) | 93.69 (0.02) | 93.36 (0.19) | 91.96 (0.06) | 86.01 (0.34) | 82.62 (0.05) |
| $L_1$ + AI (Lee et al., 2020) | 93.72 (0.18) | 93.24 (0.09) | 92.18 (0.14) | 86.67 (0.08) | 83.34 (0.18) |
| $L_1$ + KernOrth (Xie et al., 2017) | 93.49 (0.04) | 93.30 (0.19) | 91.71 (0.14) | 84.78 (0.34) | 80.87 (0.47) |
| $L_1$ + OrthConv (Wang et al., 2020) | 92.54 (0.09) | 92.41 (0.07) | 91.02 (0.16) | 84.52 (0.27) | 80.23 (1.19) |
| KernOrth (Xie et al., 2017) + $L_1$ | 93.49 (0.07) | 92.82 (0.10) | 90.54 (0.25) | 85.47 (0.20) | 79.48 (0.81) |
| OrthConv (Wang et al., 2020) + $L_1$ | 93.63 (0.17) | 93.28 (0.20) | 92.27 (0.13) | 86.70 (0.07) | 83.21 (0.61) |
| **OPP** (ours) | **93.99** (0.18) | 93.44 (0.05) | **92.43** (0.20) | **89.50** (0.08) | **85.94** (0.09) |
| | Initial | finetuning | LR 1e-3 | | |
| $L_1$ (Li et al., 2017) | 93.43 (0.06) | 93.12 (0.10) | 91.77 (0.11) | 87.57 (0.09) | 83.10 (0.12) |
| **OPP** (ours) | **93.54** (0.08) | **93.32** (0.11) | **92.00** (0.08) | **89.09** (0.10) | **85.47** (0.22) |
| Acc. diff. ($L_1$) | 0.36 | 0.39 | 0.49 | 1.14 | 1.53 |
| Acc. diff. (OPP) | 0.45 | 0.12 | 0.43 | 0.41 | 0.47 |
| **VGG19 + CIFAR100**: Baseline accuracy 74.02%, Params: 20.08M, FLOPs: 0.80G | | | | | |
| Pruning ratio $r$ | 0.1 | 0.3 | 0.5 | 0.7 | 0.9 |
| Sparsity/Speedup | 19.24%/1.23× | 51.01%/1.97× | 74.87%/3.60× | 90.98%/8.84× | 98.96%/44.22× |
| | Initial | finetuning | LR 1e-2 | | |
| Scratch | 72.84 (0.25) | 71.88 (0.14) | 70.79 (0.08) | 66.51 (0.11) | 54.37 (0.40) |
| $L_1$ (Li et al., 2017) | 74.01 (0.18) | 73.01 (0.22) | 71.49 (0.14) | 66.05 (0.04) | 51.36 (0.11) |
| $L_1$ + OrthP (Wang et al., 2021a) | 74.00 (0.04) | 72.30 (0.49) | 68.00 (0.24) | 62.22 (0.15) | 48.07 (0.31) |
| $L_1$ + AI (Lee et al., 2020) | 74.05 (0.12) | **73.08** (0.26) | 71.12 (0.23) | 64.91 (0.23) | 49.79 (0.11) |
| $L_1$ + KernOrth (Xie et al., 2017) | 73.72 (0.26) | 72.53 (0.09) | 71.23 (0.10) | 65.90 (0.14) | 50.75 (0.30) |
| $L_1$ + OrthConv (Wang et al., 2020) | 73.18 (0.10) | 72.25 (0.31) | 70.82 (0.11) | 64.51 (0.43) | 48.31 (0.18) |
| KernOrth (Xie et al., 2017) + $L_1$ | 73.73 (0.23) | 72.41 (0.12) | 70.31 (0.12) | 64.10 (0.19) | 50.72 (0.87) |
| OrthConv (Wang et al., 2020) + $L_1$ | 73.55 (0.18) | 72.67 (0.09) | 71.24 (0.23) | 65.66 (0.10) | 50.53 (0.46) |
| **OPP** (ours) | **74.30** (0.20) | 72.84 (0.07) | **71.71** (0.09) | **67.91** (0.18) | **57.81** (0.28) |
| | Initial | finetuning | LR 1e-3 | | |
| $L_1$ (Li et al., 2017) | 73.67 (0.05) | 72.04 (0.12) | 70.21 (0.02) | 64.72 (0.17) | 48.43 (0.44) |
| **OPP** (ours) | **73.83** (0.02) | **72.29** (0.07) | **71.16** (0.12) | **67.47** (0.17) | **56.73** (0.34) |
| Acc. diff. ($L_1$) | 0.34 | 0.97 | 1.28 | 1.33 | 2.93 |
| Acc. diff. (OPP) | 0.47 | 0.55 | 0.55 | 0.44 | 1.08 |

(2) We also compare the test accuracy *before finetuning* – from this metric, we will see how robust different methods are right after weight removal. This metric is also used by previous pruning works (*e.g.*, (Wang et al., 2019a)) and broadly utilized to derive advanced weight importance criteria based on Hessian (LeCun et al., 1990; Hassibi & Stork, 1993; Singh & Alistarh, 2020).

## 4.1 ANALYSIS WITH MLP-7-LINEAR ON MNIST

MLP-7-Linear is a seven-layer linear MLP. It is adopted in (Wang et al., 2021a) for analysis because linear MLP is the only network that can achieve *exact* dynamical isometry (all JSVs are exactly 1) so far. Their proposed dynamical isometry recovery method OrthP (Wang et al., 2021a) was shown can achieve exact isometry on linear MLP networks. Since we claim our method OPP can maintain dynamical isometry too, conceivably, our method should play a similar role to OrthP in pruning. To confirm this, we prune the MLP-7-Linear network with our method (exactly following the settings in (Wang et al., 2021a) for fair comparison).

Results are presented in Fig. 2, where we plot the mean JSVs and test accuracies. Mean JSV is a metric to measure dynamical isometry (Lee et al., 2020; Wang et al., 2021a). Fig. 2(b) is the one equipped with OrthP, which can exactly recover dynamical isometry (note its mean JSV right after pruning is 1.0000), so it works as the oracle here. We have the following observations. **(1)** OrthP

Table 2: Speedup comparison on ImageNet. FLOPs: ResNet34: 3.66G, ResNet50: 4.09G.

| Method | Network | Base top-1 (%) | Pruned top-1 (%) | Top-1 drop (%) | Speedup |
|---|---|---|---|---|---|
| $L_1$ (pruned-B) (Li et al., 2017) | | 73.23 | 72.17 | 1.06 | **1.32×** |
| $L_1$ (pruned-B, reimpl.) (Wang et al., 2021a) | | 73.31 | 73.67 | -0.36 | **1.32×** |
| Taylor-FO (Molchanov et al., 2019) | ResNet34 | 73.31 | 72.83 | 0.48 | 1.29× |
| GReg-2 (Wang et al., 2021c) | | 73.31 | 73.61 | -0.30 | **1.32×** |
| **OPP** (ours) | | 73.31 | **73.71** | **-0.40** | **1.32×** |
| ProvableFP (Liebenwein et al., 2020) | | 76.13 | 75.21 | 0.92 | 1.43× |
| GReg-1 (Wang et al., 2021c) | ResNet50 | 76.13 | 76.27 | -0.14 | **1.49×** |
| **OPP** (ours) | | 76.13 | **76.40** | **-0.27** | **1.49×** |
| IncReg (Wang et al., 2019b) | | 75.60 | 72.47 | 3.13 | 2.00× |
| SFP (He et al., 2018a) | | 76.15 | 74.61 | 1.54 | 1.72× |
| HRank (Lin et al., 2020) | | 76.15 | 74.98 | 1.17 | 1.78× |
| Taylor-FO (Molchanov et al., 2019) | | 76.18 | 74.50 | 1.68 | 1.82× |
| Factorized (Li et al., 2019) | | 76.15 | 74.55 | 1.60 | **2.33×** |
| DCP (Zhuang et al., 2018) | ResNet50 | 76.01 | 74.95 | 1.06 | 2.25× |
| CCP-AC (Peng et al., 2019) | | 76.15 | 75.32 | 0.83 | 2.18× |
| C-SGD-50 (Ding et al., 2019) | | 75.34 | 74.54 | 0.80 | 2.26× |
| GReg-2 (Wang et al., 2021c) | | 76.13 | 75.36 | 0.77 | 2.31× |
| CC (Li et al., 2021) | | 76.15 | 75.59 | 0.56 | 2.12× |
| **OPP** (ours) | | 76.13 | **75.60** | **0.53** | 2.31× |
| LFPC (He et al., 2020) | | 76.15 | 74.46 | 1.69 | 2.55× |
| GReg-2 (Wang et al., 2021c) | ResNet50 | 76.13 | 74.93 | 1.20 | 2.56× |
| CC (Li et al., 2021) | | 76.15 | 74.54 | 1.61 | **2.68×** |
| **OPP** (ours) | | 76.13 | **75.05** | **1.08** | 2.56× |
| IncReg (Wang et al., 2019b) | | 75.60 | 71.07 | 4.53 | 3.00× |
| Taylor-FO (Molchanov et al., 2019) | ResNet50 | 76.18 | 71.69 | 4.49 | 3.05× |
| GReg-2 (Wang et al., 2021c) | | 76.13 | 73.90 | 2.23 | **3.06×** |
| **OPP** (ours) | | 76.13 | **74.41** | **1.52** | **3.06×** |
| MobileNetV2-0.75x (Sandler et al., 2018) | | 71.88 | 69.80 | 2.08 | 1.36× |
| AMC (He et al., 2018b) | MobileNetV2 | 71.88 | 70.80 | 1.08 | 1.36× |
| CC (Li et al., 2021) | | 71.88 | 70.91 | 0.97 | 1.40× |
| **OPP** (ours) | | 71.88 | **71.01** | **0.87** | **1.41×** |

improves the best accuracy from 91.36/90.54 to 92.79/92.77; using OPP, we obtain 92.81/92.77. Namely, in terms of accuracy, our method is *as good as the oracle scheme*, if not better. **(2)** Note the mean JSV right after pruning – the $L_1$-norm pruning destroys the mean JSV from 2.4987 to 0.0040, and OrthP brings it back to 1.0000. In comparison, OPP achieves 3.4875, also comparable to OrthP.

## 4.2 RESNET56+CIFAR10 / VGG19+CIFAR100

Here we compare our method to other plausible solutions on the CIFAR10/100 datasets (Krizhevsky, 2009) with non-linear convolutional networks (ResNet56 and VGG19). From the results in Tab. 1, we have the following observations.

(1) OrthP (Wang et al., 2021a) and AI (Lee et al., 2020) do not work well – $L_1$+OrthP underperform the original $L_1$ under all the five pruning ratios for both ResNet56 and VGG19. $L_1$+AI also underperforms $L_1$ on ResNet56 for all pruning ratios. On VGG19, $L_1$+AI only works for small pruning ratio regime (such as $r = 0.1$ and $r = 0.3$).

(2) For KernOrth vs. OrthConv, the results look mixed – OrthConv is better when applied *before* the $L_1$-norm pruning. This is reasonable since OrthConv has been shown more effective than KernOrth in enforcing more dynamical isometry (Wang et al., 2020), which in turn can stand more of the damage from pruning.

(3) Of special note is that, none of the above 6 methods actually outperform the $L_1$-norm pruning or the simple scratch training in aggressive pruning cases (the last two columns). It means that neither enforcing more isometry before pruning nor compensating isometry after pruning can help dynamical isometry recovery. In stark contrast, our proposed OPP method outperforms $L_1$-norm pruning and scratch *consistently* under most pruning ratios, especially in large pruning ratio regime. Besides, note that the accuracy trend: in general, with a *larger* pruning ratio, the advantage of OPP over $L_1$ or Scratch is *more pronounced*. This is because a larger pruning ratio means the dynamical isometry is damaged more, where our method can help more, thus harvest more performance gains.

Table 3: Ablation Study: Test accuracy (without finetuning) comparison with or without the proposed BN regularization.

| ResNet56 + CIFAR10: Baseline accuracy 93.78%, Params: 0.85M, FLOPs: 0.25G | | | | | |
|---|---|---|---|---|---|
| Pruning ratio $r$ | 0.3 | 0.5 | 0.7 | 0.9 | 0.95 |
| OPP (w/o BN reg) | 92.79 (0.03) | 92.23 (0.08) | 90.46 (0.21) | 44.25 (2.46) | 16.52 (0.43) |
| OPP (w/ BN reg) | **92.94** (0.14) | **92.48** (0.19) | **90.48** (0.09) | **70.53** (1.69) | **23.05** (2.61) |
| Acc. diff. | +0.15 | +0.25 | +0.02 | +26.28 | +6.53 |
| VGG19 + CIFAR100: Baseline accuracy 74.02%, Params: 20.08M, FLOPs: 0.80G | | | | | |
| Pruning ratio $r$ | 0.1 | 0.3 | 0.5 | 0.7 | 0.9 |
| OPP (w/o BN reg) | 73.01 (0.13) | 71.26 (0.19) | 68.67 (0.10) | 61.70 (0.46) | 1.75 (0.38) |
| OPP (w/ BN reg) | **73.44** (0.07) | **71.61** (0.12) | **69.28** (0.25) | **65.15** (0.20) | **2.84** (1.13) |
| Acc. diff. | +0.43 | +0.35 | +0.51 | +3.45 | +1.09 |

(4) In Tab. 1, we also present the results when the initial finetuning LR is `1e-3`. In (Wang et al., 2021a), is is shown that if the broken dynamical isometry can be well maintained/recovered, the final performance gap between LR `1e-2` and `1e-3` will be diminished. Since we claim our method is good at maintaining dynamical isometry, the performance gap should become smaller. This is empirically verified in the table (note the Acc. diff. of OPP vs. that of $L_1$). In general, the accuracy gap between LR `1e-2` and LR `1e-3` of OPP is smaller than that of $L_1$-norm pruning. Two exceptions are $r = 0.3$ for ResNet56 and $r = 0.1$ for VGG9. We conceive this is because the speedup ratio is small. The accuracy difference between LR `1e-2` and `1e-3` is small itself. Despite them, the general picture from the table is that the accuracy gap between LR `1e-3` and `1e-2` turns smaller with our method, which is a good sign that dynamical isometry is effectively maintained.

### 4.3 IMAGENET

We further evaluate OPP on ImageNet (Deng et al., 2009) in comparison to many existing filter pruning algorithms. Results are shown in Tab. 2. Our method is *consistently* better than the others across different speedup ratios. Moreover, with a larger speedup ratio, the advantage of our method is more pronounced. For example, OPP outperforms Taylor-FO (Molchanov et al., 2019) by 1.15% in terms of the top-1 accuracy drop at the 2.31×speedup track; at 3.06×speedup, OPP leads Taylor-FO (Molchanov et al., 2019) by 2.87%. This shows OPP is more robust to more aggressive pruning. The reason is easy to see – more aggressive pruning hurts dynamical isometry more (Lee et al., 2019; Wang et al., 2021a). Our method can find more use in these cases since it can recover the broken dynamical isometry during pruning. This phenomenon has been observed consistently (see also the results in Tab. 1 and ablation study in Tab. 3).

### 4.4 ABLATION STUDY

Finally, we conduct ablation study regarding regularizing the two learnable parameters in BN layers. The results are presented in Tab. 3, where we compare the accuracy right after pruning (namely, without finetuning). As seen, when the BN regularization is switched off, the performance degrades. A clear trend is: BN regularization is *more helpful* under the *larger sparsity*. This reiterates the importance of regularizing BN for dynamical isometry recovery, especially in the aggressive pruning.

## 5 CONCLUSION

Dynamical isometry maintenance or recovery is crucial to neural network pruning. In this work, we present a dynamical isometry preserving pruning method (named orthogonality preserving pruning) based on regularization. Specifically, we propose a modified weight gram matrix as regularization target which drives the unimportant weights towards zero meanwhile regularizing the remaining ones to archive orthogonality. Besides, we propose to regularize the BN parameters to mitigate its damage to dynamical isometry. The proposed method is shown much more effective than existing isometry recovery counterparts. The resulted pruning algorithm also performs better compared to many recent filter pruning approaches. To our best knowledge, this is the *first* dynamical isometry preserving filter pruning method that scales to the large-scale ImageNet dataset.

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

# A  IMPLEMENTATION DETAILS

## A.1  CODE REFERENCE

We mainly refer to the following code implementations in this work. They are all open-licensed.

- Official PyTorch ImageNet example: https://github.com/pytorch/examples/tree/master/imagenet;
- GReg-1/GReg-2 (Wang et al., 2021c): https://github.com/MingSun-Tse/Regularization-Pruning;
- OrthConv (Wang et al., 2020): https://github.com/samaonline/Orthogonal-Convolutional-Neural-Networks;
- Rethinking the value of network pruning (Liu et al., 2019): https://github.com/Eric-mingjie/rethinking-network-pruning/tree/master/imagenet/l1-norm-pruning.

**Data split**. All the datasets in this paper are public datasets with standard APIs in PyTorch (Paszke et al., 2019). We employs these standard APIs for the train/test data split to keep fair comparison with other methods.

## A.2  TRAINING SETTING

The specific training settings are summarized in Tab. 4. For the hyper-parameters that are introduced in our OPP method: regularization granularity $\Delta$, regularization ceiling $\tau$ and regularization update interval $K_u$, we summarize them in Tab. 5. These hyper-parameters mainly refer to the official implementation of GReg-1 (Wang et al., 2021c) since we tap into a similar growing regularization scheme as it does.

Table 4: Training setting summary. For the SGD solver, in the parentheses are the momentum and weight decay. For LR schedule, the first number is initial LR; the second (in brackets) is the epochs when LR is decayed by factor 1/10; and #epochs stands for the total number of epochs.

| Dataset | MNIST | CIFAR10/100 | ImageNet |
|---|---|---|---|
| Solver | SGD (0.9, 1e-4) | SGD (0.9, 5e-4) | SGD (0.9, 1e-4) |
| Batch size | 100 | CIFAR10: 128; CIFAR100, ImageNet: 256 | |
| LR schedule (scratch) | 1e-2, [30,60], #epochs:90 | 1e-1, [100,150], #epochs:200 | 1e-1, [30,60], #epochs:90 |
| LR schedule (prune) | Fixed (1e-3) | | |
| LR schedule (finetune) | 1e-2, [30,60], #epochs:90 | 1e-2, [60,90], #epochs:120 | 1e-2, [30,60,75], #epochs:90 |

Table 5: Hyper-parameters of our methods.

| Dataset | MNIST | CIFAR10/100 | ImageNet |
|---|---|---|---|
| Regularization granularity $\Delta$ | | 1e-4 | |
| Regularization ceiling $\tau$ | | 1 | |
| Regularization update interval $K_u$ | | 10 iterations | 5 iterations |

## A.3  HARDWARE AND RUNNING TIME

We conduct all our experiments using 4 NVIDIA V100 GPUs (16GB memory per GPU). It takes roughly 41 hrs to prune ResNet50 on ImageNet using our OPP method (pruning and 90-epoch finetuning both included). Among them, 12 hrs (namely, close to 30%) are spent on pruning and 29 hrs are spent on finetuning (about 20 mins per epoch).

# B  LAYER-WISE PRUNING RATIOS

The layer-wise pruning ratios are pre-specified in this paper. For the ImageNet benchmark, we *exactly* follow GReg-1/GReg-2 (Wang et al., 2021c) for the layer-wise pruning ratios to keep fair comparison to it. The specific numbers are summarized in Tab. 6. Each number is the pruning ratio shared by all the layers of a stage in ResNet34/50.

**Reason for using pre-defined pruning ratios**. In this paper consider pruning independent with the pruning ratio choosing. The main consideration is that pruning ratio is broadly believed to reflect

Table 6: Summary of layer-wise pruning ratios.

| Dataset | Network | Speedup | Pruned top-1 accuracy (%) | Pruning ratio |
|---------|---------|---------|---------------------------|---------------|
| ImageNet | ResNet34 | $1.32\times$ | 73.77 | [0, 0.50, 0.60, 0.40, 0, 0]* |
| ImageNet | ResNet50 | $1.49\times$ | 76.44 | [0, 0.30, 0.30, 0.30, 0.14, 0] |
| ImageNet | ResNet50 | $2.31\times$ | 75.60 | [0, 0.60, 0.60, 0.60, 0.21, 0] |
| ImageNet | ResNet50 | $2.56\times$ | 75.12 | [0, 0.74, 0.74, 0.60, 0.21, 0] |
| ImageNet | ResNet50 | $3.06\times$ | 74.51 | [0, 0.68, 0.68, 0.68, 0.50, 0] |

*In addition to the pruning ratios, we skip several layers, following the setting of $L_1$ (pruned-B) (Li et al., 2017) for fair comparison. Concretely, we refer to the implementation of (Liu et al., 2019) at https://github.com/Eric-mingjie/rethinking-network-pruning/tree/master/imagenet/l1-norm-pruning.

the redundancy of different layers, which is an inherent characteristic of the model, thus should not be coupled with the subsequent pruning algorithms. Thus, we choose to set pruning ratios before pruning. Exploring a better pruning ratio is not our focus in this paper given the limited length.

In fact, showing the effectiveness of the pruning ratios is unnecessary to our paper as long as we use the same ratios as GReg-1/2 does to keep a fair comparison. The reason is, if the pruning ratios we used turns out good, there is no concern about it; if they turn out bad, that's even better, because it shows our method can achieve SOTA performance even if using such a bad pruning ratio combination. Either case, it does not undermine the value of our method.

## C SENSITIVITY ANALYSIS OF HYPER-PARAMETERS

Among the three hyper-parameters in Tab. 5, regularization ceiling $\tau$ works as a termination condition. We only requires it to be large enough to ensure the weights are compressed to a very small amount. It does not have to be 1. The final performance is also less sensitive to it. The pruned performance seems to be more sensitive to the other two hyper-parameters, so here we conduct hyper-parameter sensitivity analysis to check their robustness.

Results are presented in Tab. 7 and Tab. 8. Pruning ratio 0.7 (for ResNet56) and 0.5 (for VGG19) are chosen here because the resulted sparsity is the most typical (*i.e.*, not too large or small). (1) For $K_u$, in general, a larger $K_u$ tends to deliver a better result. This is no surprise since a larger $K_u$ implies more time for the network to adapt and recover when undergoing the penalty. (2) For $\Delta$, we do not see a clear pattern here: either a small or large $\Delta$ can achieve the best result (for different networks). On the whole, when varying the hyper-parameters within a reasonable range, the performance is pretty robust, no catastrophic failures. Moreover, note that the default setting is actually not the best for both $K_u$ and $\Delta$. This is because we did *not* hard-tune our hyper-parameters, however, they still achieve encouraging performance compared to the counterpart methods, as we have shown in the main text.

Table 7: Sensitivity analysis of $K_u$ on CIFAR10/100 datasets with the proposed OPP algorithm. $K_u = 10$ is the default setting. Pruning ratio 0.7 (ResNet56) and 0.5 (VGG19) are explored here. Each setting is randomly run for 3 times, mean and standard deviation accuracy reported. **The best is highlighted with bold** and the worst is highlighted with blue color.

| $K_u$ | 1 | 5 | 10 | 15 | 20 |
|-------|---|---|----|----|----|
| Acc. (%, ResNet56) | $92.24_{\pm 0.03}$ | $92.42_{\pm 0.14}$ | $92.43_{\pm 0.20}$ | $\mathbf{92.50_{\pm 0.10}}$ | $92.31_{\pm 0.16}$ |
| Acc. (%, VGG19) | $71.33_{\pm 0.06}$ | $71.45_{\pm 0.21}$ | $\mathbf{71.71_{\pm 0.09}}$ | $71.43_{\pm 0.21}$ | $71.68_{\pm 0.18}$ |

Table 8: Sensitivity analysis of $\Delta$ on CIFAR10/100 datasets with the proposed OPP algorithm. $\Delta = 1e-4$ is the default setting. Pruning ratio 0.7 (ResNet56) and 0.5 (VGG19) are explored here. Each setting is randomly run for 3 times, mean and standard deviation accuracy reported. **The best is highlighted with bold** and the worst is highlighted with blue color.

| $\Delta$ | 1e-5 | 5e-5 | 1e-4 | 5e-4 | 1e-3 |
|----------|------|------|------|------|------|
| Acc. (%, ResNet56) | $92.37_{\pm 0.12}$ | $92.29_{\pm 0.10}$ | $92.43_{\pm 0.20}$ | $92.40_{\pm 0.15}$ | $\mathbf{92.44_{\pm 0.13}}$ |
| Acc. (%, VGG19) | $71.39_{\pm 0.19}$ | $71.37_{\pm 0.14}$ | $\mathbf{71.71_{\pm 0.09}}$ | $71.58_{\pm 0.31}$ | $71.25_{\pm 0.31}$ |

# D    MORE BACKGROUND OF DYNAMICAL ISOMETRY

In the main paper (Fig. 2), we present the mean JSV (0.0040) right after pruning and those after using OrthP (1.0000) and our OPP method (3.4875). We say "OPP achieves 3.4875, also comparable to OrthP". How to understand this "comparable"? By what standards? This section is meant to provide more background concerning these questions.

Dynamical isometry (DI) is defined by mean JSV close to 1 in (Saxe et al., 2014) (rigorously in (Saxe et al., 2014), DI describes the distribution of all JSVs. Mean JSV is only an average sketch of the distribution. But this approximation is accurate enough for analysis in this paper, so we say "DI is defined by mean JSV"). In other words, if a network has mean JSV close to 1, we can say this network has dynamical isometry. This is a fundamental basis of our paper.

Then, a non-trivial technical question we need to answer is: When we deal with practical deep neural networks in the real world, **how close is the so-called "close to 1" in the above definition**? To our best knowledge, there is no outstanding theory to quantify this, so we resort to empirical analysis. In Tab. 9 below, we present the mean JSV of MLP-7-Linear network on MNIST under different pruning ratios, along with their accuracies before and after finetuning.

Table 9: Mean JSV and test accuracies (%) of MLP-7-Linear on MNIST under different pruning ratios. Each result is randomly run for 3 times. We report the mean accuracy and (std). "ft." is short for finetuning.

| Pruning ratio | 0 | 0.1 | 0.2 | 0.3 | 0.4 | 0.5 | 0.6 | 0.7 | 0.8 | 0.9 |
|---|---|---|---|---|---|---|---|---|---|---|
| mean JSV | 2.4987 | 1.7132 | 0.9993 | 0.5325 | 0.2711 | 0.1180 | 0.0452 | 0.0151 | 0.0040 | 0.0004 |
| Acc. before ft. | 92.77 | 91.35 | 78.88 | 62.21 | 32.14 | 11.47 | 9.74 | 9.74 | 9.74 | 9.74 |
| Acc. after ft. | / | 92.82 (0.05) | 92.80 (0.04) | 92.80 (0.01) | 92.77 (0.01) | 92.77 (0.02) | 92.77 (0.00) | 92.78 (0.02) | 91.37 (0.03) | 87.82 (0.03) |

As seen, there is a clear trend – larger pruning ratio, lower accuracy (before or after finetuning), smaller mean JSV. Of special note is the mean JSV range where the pruned network can be *finetuned back to the original accuracy* (92.77%) – [0.0151, 2.4987]. This means, for networks with mean JSV from 0.0151 to 2.4987, in spite that their immediate accuracies (without finetuning) can be distinct (e.g., 91.35% vs. 9.74%), *intrinsically, they are equivalently potential after finetuning*. By this "equivalently potential" rule, all the mean JSVs in the range of [0.0151, 2.4987] can be regarded as comparable (although the two extremes have a gap of two orders of magnitude). That is, two networks, as long as their accuracies after finetuning are comparable, we deem their mean JSVs as comparable.

In Fig. 2, OPP achieves 92.87/92.77 test accuracy, as comparable to 92.79/92.77 when using OrthP. Also, 3.4875 is close to 2.4987 in the same order of magnitude. Thus, we say OPP is comparable to OrthP in Fig. 2.

