# OpenReview forum: "Structured Pruning Meets Orthogonality"
_ICLR.cc/2022/Conference — ICLR 2022 Submitted_

### Official Review · Reviewer_idn1 · 2021-10-31

**Correctness:** 2
**Technical Novelty And Significance:** 2
**Empirical Novelty And Significance:** 2
**Recommendation:** 3
**Confidence:** 5

**Main Review:**

The paper is well written in general, and the proposed methods are intuitive. In the experimental analyses, the proposed methods outperform baseline on various benchmark datasets.

However, there are few major issues with the paper:

- There are three main claims in the paper; achieving structured pruning, orthogonality and dynamic isometry. However, these claims are not explored in detail;

-- Structured pruning is only mentioned as a synonym for filter pruning or channel pruning in the review of the related work. This is achieved in the proposed method by user defined parameters that identify the indices of parameters to be pruned. In the analyses, selection of this set of indices affects the accuracy as well. That is, there is not a major contribution for structured pruning.

-- Orthogonality is aimed to be achieved by regularization on gram matrices of kernels. However, how well/much orthogonalization is obtained is not analyzed. In addition, regularization of batch normalization parameters has much more effect on the accuracy compared to orthogonalization.

-- The paper claims to achieve dynamic isometry by orht. However, it is also stated “We also propose to regularize batch-normalization parameters for better preserving dynamical isometry for the whole network.” That is, either orth. does not enable dynamic isometry by itself, or dynamic isometry does not affect the accuracy remarkably. This claim should be examined theoretically or experimentally in detail.

To summarize, the relationship between these three proposed major claims should be analyzed in detail. Otherwise, the scope and claims of the paper should be updated accordingly.

Additional questions:

-	How did you calculate speedup? Could you please provide an example of actual running times for training and test for an experiments, such for the VGG19 + Cifar100 case, in comparison with the baseline?

-	Have you employed regularization of parameters of BN of baseline methods to check how this method also affects accuracy of baseline?


**Summary Of The Paper:**

This paper proposes regularizing learnable parameters of NNs to maintain the dynamical isometry during pruning and improve their accuracy. In the experimental analyses, the proposed OPP outperforms baseline methods on benchmark datasets.

The paper is well written in general, and the proposed methods are intuitive. In the experimental analyses, the proposed methods outperform baseline on various benchmark datasets.


**Summary Of The Review:**

The paper is well written in general, and the proposed methods are intuitive. However, the proposed claims are not explored well. More precisely, the relationship between the proposed major claims should be analyzed in detail. Otherwise, the scope and claims of the paper should be updated accordingly. To this end, the paper proposes regularizing learnable parameters of NNs using different heuristics. Therefore, the novelty and scope can be redefined according to provided analyses and results.

---

> ### Author Response · Authors · 2021-11-22
> **Responses to Reviewer idn1**
>
> We greatly thank Reviewer idn1 (R4) for your constructive comments! Your concerns are addressed below.
>
> `R4-Q1`: Structured pruning is only mentioned as a synonym for filter pruning or channel pruning in the review of the related work. This is achieved in the proposed method by user defined parameters that identify the indices of parameters to be pruned. In the analyses, selection of this set of indices affects the accuracy as well. That is, there is not a major contribution for structured pruning.
>
> `R4-A1`: Your understanding is correct! As we stated in the paper "we sort the filters by their L1-norms and select those with the least L1-norms as unimportant filters", we select the unimportant filters just like L1-norm pruning (Li et al., 2017). We actually did *not* claim any contribution from this side. The major contribution of this paper is that, structured pruning (or filter pruning in the context of this paper) was shown breaking dynamical isometry. Previous papers have not addressed this problem, while our method helps resolve this issue for the first time on *large-scale non-linear convolutional* networks.
>
> `R4-Q2`: Orthogonality is aimed to be achieved by regularization on gram matrices of kernels. However, how well/much orthogonalization is obtained is not analyzed. In addition, regularization of batch normalization parameters has much more effect on the accuracy compared to orthogonalization.
>
> `R4-A2`: Orthogonality is a feasible tool to achieve dynamical isometry. The final end should be dynamical isometry. Thus, "how well/much orthogonalization is obtained" actually reduces to "how well/much dynamical isometry is obtained". In this sense, mean JSV is employed as the metric for dynamical isometry. We already provide the specific mean JSV values on MLP-7-Linear for analysis in Fig. 2 (e.g., OPP improves the mean JSV from 0.0040 to 3.4875), also analyses in Sec. 4.1.
>
> `R4-Q3`: The paper claims to achieve dynamic isometry by orht. However, it is also stated “We also propose to regularize batch-normalization parameters for better preserving dynamical isometry for the whole network.” That is, either orth. does not enable dynamic isometry by itself, or dynamic isometry does not affect the accuracy remarkably. This claim should be examined theoretically or experimentally in detail.
>
> `R4-A3`: Thanks to R4 for pointing out this critical concept is not clearly stated in our paper. When we say "achieving dynamic isometry by orthogonalization", what we really mean is "*approximately* achieving dynamical isometry". Exact dynamical isometry can only be achieved via orthogonalization on *Linear MLP* networks. Since we are dealing with *non-linear convolutional* network, orthogonalization can only achieve dynamical isometry *approximately*. Therefore, R4's statement "orth. does not enable dynamic isometry by itself" is more correct. Yet please note, *this is not a problem that originates from this paper, but a general open question in the area of dynamical isometry*. We'll make the statements more rigorous to avoid such confusion.
>
> `R4-Q4`: How did you calculate speedup?
>
> `R4-A4`: Speedup is calculated by FLOPs (a.k.a. Mult-Adds) reduction. For example, if an unpruned network has FLOPs $10$, we reduce the FLOPs to $4$, then the speedup is $10/4=2.5 \times$. This is the most standard way to calculate speedup in the pruning community. We simply follow the common practice.
>
> `R4-Q5`: Have you employed regularization of parameters of BN of baseline methods to check how this method also affects accuracy of baseline?
>
> `R4-A5`: The BN regularization is a part of *our method*, which may not be naturally compatible with other baseline methods. For example, L1-norm pruning is a *one-shot* pruning method. It does not have a training process for pruning, thus cannot be integrated with BN regularization. Actually, we are not very clear about which "baseline methods" R4 is referring to here. You may suggest specific method names and see if we can help resolve your concern.
>
> Hope our feedback can help resolve your concerns! *If you have more regarding this feedback, please let us know!*

---

> ### Author Response · Authors · 2021-11-30
> **Sincerely expecting feedback from Reviewer idn1**
>
> Dear Reviewer idn1,
>
> We greatly thank you for the reviewing process so far! Although the ICLR final discussion deadline is (extremely) approaching, we *really* hope to have some feedback from you to see if our responses resolve your concerns. Thank you very much!!
>
> Sincerely,
>
> Authors

---

> ### Comment · Reviewer_idn1 · 2021-11-30
> **Additional feedback.**
>
> Dear Authors,
>
> Thank you for the responses. I checked all the reviewer comments and author responses, together with the paper several times. A few additional comments are as follows:
>
> Regarding isometry: There are two major issues;
>
> 1) The paper states that “Mean JSV is only an average sketch of the distribution. But this approximation is accurate enough for analysis in this paper, so we say “DI is defined by mean JSV”). In other words, if a network has mean JSV close to 1, we can say this network has dynamical isometry. This is a fundamental basis of our paper.”  First, you should not that Saxe et al. propose exact solutions to the nonlinear dynamics of learning in deep linear neural networks. Then, they discuss about approximation for nonlinear networks. This is also addressed by “Therefore, R4's statement "orth. does not enable dynamic isometry by itself" is more correct. Yet please note, this is not a problem that originates from this paper, but a general open question in the area of dynamical isometry.” However, as also stated in the paper, this is the fundamental claim and basis of the whole paper. Therefore, you need to provide more precise results for nonlinear networks to support your fundamental basis, or revise the basis.
>
> 2)  In the interpretation of experimental results, it is stated that “JSV 1 is the definition of exact isometry, which is the ideal case. In a practical DNN, JSV 1 rarely happens (unless we manually set it to 1).”.  This contradictory results may be due to the approximation issue occurring for nonlinear networks as mentioned above. In addition, this is one of the main claims and basis of the proposed method. Therefore, this issue should be clarified or the claims should be revised.
>
> Regarding structured pruning: Please directly refer to filter pruning instead of using a more general collection of structured pruning approaches by revising the title and other related novelty/contributions in the paper. Please also define the pruned filters more precisely. In the community, some papers consider a filter by an InxWxH tensor, where In is the number of input channels, W and H denote width and height of the kernel, while the others define a filter by an out x In.W.H matrix where out denotes number of output channels and In.W.H is the scalar multiplication of In, W and H. In your paper, you also prune networks with different structures such as ResNets (with filters in the residual connections) and VGG. Then, how do you define a filter to be pruned more precisely?
>
> - Also, what does \bigdot in eqn (3) denote?

---

> > ### Author Response · Authors · 2021-11-30
> > **Thanks For Your Further Comments**
> >
> > Dear Reviewer idn1 (R4),
> >
> > Terrific to see your feedback. We address your further concerns as follows.
> >
> > 1. "*you need to provide more precise results for nonlinear networks to support your fundamental basis, or revise the basis.*" -- This is still a questioning regarding why we use the (possibly flawed) mean JSV metric for dynamical isometry. We have the following reasons. (1) Mean JSV, admittedly, is not a perfect metric for dynamical isometry. However, we use it not because it is the best, but because we *do not have a better alternative* (if R4 disagrees, you may specifically suggest another one). R4 requires us to "provide more precise results for nonlinear networks to support your fundamental basis", which may be over-demanding in our view, because dynamical isometry for non-linear networks itself is still an open (and hard) problem in the area.  (2) Also importantly, this metric *has been successfully used by previous works* (such as (Lee et al., 2020)). If they can use this metric for analysis, we do not see why we cannot use it here, especially when we actually do not have another better choice. (3) The metric may not be perfect, but is still *very informative and pertinent* to our analysis, as we've shown in Appendix D.
> >
> > Given these facts, we believe it is fair for us to use mean JSV as a metric for analysis and sincerely hope R4 can agree with this.
> >
> > 2. "*“JSV 1 is the definition of exact isometry, which is the ideal case. In a practical DNN, JSV 1 rarely happens (unless we manually set it to 1).”. This contradictory results may be due to the approximation issue occurring for nonlinear networks as mentioned above.*" -- Yes, this is due to the non-linearity in a practical network. Here we just point out we cannot demand the mean JSV to be exactly 1 for a practical DNN (to be called *isometry*). Otherwise, no further analysis can be done since mean JSV = 1 only happens for the initialization of a network. When it starts training, mean JSV definitely will deviate from 1. We'll add these discussions in the paper and make it more clear. Yet, it may not be considered as an issue/weakness (at least for this paper). Again -- addressing the dynamical isometry for non-linear networks or networks during training instead of initialization is *not* the focus/responsibility/motivation of this paper. We are using the state-of-the-art tool from the area of dynamical isometry to *study filter pruning*. As for the issue that the tool itself is not perfect, we believe it should not be considered as a weakness of *our* paper. We are already using the best tool available and other works (e.g., Lee et al., 2020) also used it.
> >
> > 3. "*how do you define a filter to be pruned more precisely?*" -- Thanks for pointing this out. We fully agree with the concept difference between structured pruning vs. filter pruning and will revise our title/claims to filter pruning.
> > We define filter as a 3D tensor in a convolutional layer. E.g., for a typical conv layer, parameterized by a 4D tensor of shape [N, C, H, W], the 3D tensor [C, H, W] is a filter in our paper.
> >
> > 4. "*Also, what does \bigdot in eqn (3) denote?*" -- This is a typo, also pointed out by R2 (ipz6). We will correct this and fully proofread our paper more carefully.
> >
> > Before this rebuttal window really closes, we are more than happy to take more questions. Meanwhile, if our responses so far help R4 clear (some of) your major concerns, we sincerely hope you could kindly raise your score slightly, even though you think this paper may not be good enough for publication, as it would be a fair and encouraging recognition to our efforts so far. Thanks!
> >
> > Sincerely,
> >
> > Authors

---

### Official Review · Reviewer_8kct · 2021-11-02

**Correctness:** 3
**Technical Novelty And Significance:** 2
**Empirical Novelty And Significance:** 2
**Recommendation:** 3
**Confidence:** 4

**Main Review:**

Pros:
1. Clear motivation and writing.
2. Well-established pipeline.

Cons:
1. Comparisons with the state-of-the-art methods are insufficient, such as [1].
2. Experiments on ImageNet are insufficient and results are not comparable.
3. Moderate novelty. This kernel orthogonality idea has been employed in deep networks in previous methods.


[1] ResRep: Lossless CNN Pruning via Decoupling Remembering and Forgetting, Ding et al., ICCV 2021

**Summary Of The Paper:**

This paper proposes a structured pruning method based on kernel orthogonality analysis. The method is straightforward and well-investigated in the network pruning field.

**Summary Of The Review:**

This paper has moderate novelty with the well-studied kernel orthogonality method. Besides, the experiment design is not sufficient. The performance gain in ImageNet is incremental and not sufficient compared to the state-of-the-art methods.

---

> ### Author Response · Authors · 2021-11-22
> **Responses to Reviewer 8kct (R3)**
>
> We greatly thank Reviewer 8kct (R3) for your constructive comments! Your concerns are addressed below.
>
> `R3-Q1`: Comparisons with the state-of-the-art methods are insufficient, such as [1].
>
> `R3-A1`: We are very grateful to R3 for letting us know about this related work. We shall cite this paper and include it for comparison in our new version. Meanwhile, notably, this paper is just presented in ICCV21, which is later than the ICLR22 submission date. By the common practice, we may not be discredited for not comparing our method to it.
>
> `R3-Q2`: Experiments on ImageNet are insufficient and results are not comparable.
>
> `R3-A2`: We compare with 16 methods in Tab. 2. Many of them are among the recent state-of-the-arts (such as GReg, CC, C-SGD). Two of them are recent 2021 papers (GReg: ICLR21, CC: CVPR21). We believe these are pretty much sufficient.
>
> As for "results are not comparable", could R3 be more specific with this statement so that we can address your concern? For example, which results do you think are not comparable?
>
> `R3-Q3`: Moderate novelty. This kernel orthogonality idea has been employed in deep networks in previous methods.
>
> `R3-A3`: Kernel orthogonality idea is indeed not new. This has been clearly discussed in our related work section (see "Dynamical isometry and orthogonality"). However, as far as we know, *joining structured pruning with kernel orthogonality* has not been done by any paper before (if R3 disagrees, you may suggest specific references), which is the main novelty of our paper.
>
> Hope our feedback can help resolve your concerns! *If you have more regarding this feedback, please let us know!*

---

> ### Author Response · Authors · 2021-11-30
> **Sincerely expecting feedback from Reviewer 8kct**
>
> Dear Reviewer 8kct,
>
> We greatly thank you for the reviewing process so far! Although the ICLR final discussion deadline is (extremely) approaching, we *really* hope to have some feedback from you to see if our responses resolve your concerns. Thank you very much!!
>
> Sincerely,
>
> Authors

---

> ### Author Response · Authors · 2021-12-01
> **Sincerely expecting feedback from Reviewer 8kct**
>
> Dear Reviewer 8kct,
>
> Sorry to bother you if any. We greatly thank you for the reviewing process so far! Even though you do not recommend acceptance for this paper, we *really* hope to have some feedback from you still, since *your opinions are critical* for us to improve the paper. If possible, please take a look at our responses and see if they resolve your concerns. If (some of) your major concerns are addressed, we sincerely hope you can kindly raise your score even slightly as a fair and encouraging recognition of our efforts. Thank you very much!!
>
> Sincerely,
>
> Authors

---

### Official Review · Reviewer_ipz6 · 2021-11-03

**Correctness:** 3
**Technical Novelty And Significance:** 2
**Empirical Novelty And Significance:** 2
**Recommendation:** 6
**Confidence:** 5

**Main Review:**

# Strengths

1. Orthogonality preserving structured pruning is interesting.
2. The results demonstrate the merits of the method.
3. Overall the paper is well written.

# Weaknesses

1. The role of BN regularization is not clear to me. It is not clear how the BN parameters of pruned filters would make a difference in dynamical isometry (DI) and how the proposed regularization alleviates this issue? More clarity is required.

2. It is not clear that the method indeed improves DI. Considering Fig.2 c, mean JSV increases and is higher than 1. So not clear the method indeed improves DI.

3. The notation inside the Frobenius norm is not clear in Eq.3. Probably a type, please comment.

**Summary Of The Paper:**

The paper proposes a dynamical isometry preserving approach for structured pruning by introducing an orthogonality preserving regularization for weight matrices in each layer. The experiments on cifar and imagenet datasets show improvements over comparable methods.

**Summary Of The Review:**

Overall the paper is interesting. The connection to BN regularization and DI could be explained better to improve clarity.

---

> ### Author Response · Authors · 2021-11-22
> **Responses to Reviewer ipz6 (R2)**
>
> We greatly thank Reviewer ipz6  (R2) for your constructive comments! Your concerns are addressed below.
>
> `R2-Q1`: The role of BN regularization is not clear to me. It is not clear how the BN parameters of pruned filters would make a difference in dynamical isometry (DI) and how the proposed regularization alleviates this issue? More clarity is required.
>
> `R2-A1`: DI is measured for the *whole* network. If filters in a layer are pruned, the output distribution will definitely change. Ideally, BN parameters should be adapted accordingly to account for this distribution change. For example, if the i-th filter is pruned, the i-th channel of the associated BN layer should also be removed. The point here is, the other BN parameters should be prepared for this loss so that it will not incur damage to the dynamical isometry of the whole network. The proposed regularization now achieves this goal by regularizing the BN scales and shifts to zero first, forcing the other BN parameters to be independent of them. Then, when those BN channels are physically removed, the others will not stand much loss. Hence the dynamical isometry is (fairly) maintained for the whole network.
>
> Thanks for letting us know the connection between BN regularization and DI is not clearly stated in our paper. We will add the above explanations in our paper.
>
> `R2-Q2`: It is not clear that the method indeed improves DI. Considering Fig.2 c, mean JSV increases and is higher than 1. So not clear the method indeed improves DI.
>
> `R2-A2`: Isometry is defined as JSV is close to 1. Exact isometry (JSV = 1) is the *ideal case*. For *practical* networks, it rarely happens. In Fig. 2 (c), we plot the mean JSV of a real network thus its mean JSV can be greater than 1; but this does not mean it cannot be considered as isometry -- In Appendix D, we provide an empirical study about how to look at the mean JSV *at a proper scale*. By the analysis there, the range of mean JSV in Fig. 2 can still be considered as "close to 1". In this sense, our method indeed improves DI. (If we demand DI as "JSV = 1" strictly, then probably there is no method that can improve DI in practice; note, even the OrthP method, which archives *exact isometry* in the beginning, still sees the mean JSV arising later).
>
> `R2-Q3`: The notation inside the Frobenius norm is not clear in Eq.3. Probably a type, please comment.
>
> `R2-A3`: Thanks for pointing it out! The element-wise product is a typo. We'll remove it and further proofread our paper more thoroughly.
>
> Hope our feedback can help resolve your concerns! *If you have more regarding this feedback, please let us know!*

---

> > ### Comment · Reviewer_ipz6 · 2021-11-25
> > **Questions on BN regularization**
> >
> > Thanks for the response.
> >
> > 1. I still have questions on the BN regularization. As mentioned above if the filter is pruned, one can simply remove the corresponding BN parameters. Why would one need to do this BN regularization? If the purpose is to modify the network JSV values via regularizing BN, more discussion is needed. Also, it is not clear how not regularizing BN can "damage" dynamical isometry. If I understand correctly, if a filter is pruned but the BN parameters are not removed, they have no effect on the network or its Jacobian. What am I missing?
> >
> > 2. I would recommend adding a brief discussion in the main paper mentioning in practice, there is a range of JSV values that improves performance.

---

> > > ### Author Response · Authors · 2021-11-30
> > > **Thanks For Your Further Comments!**
> > >
> > > Dear Reviewer ipz6,
> > >
> > > Great thanks for your further comments. We address your questions as follows.
> > >
> > > 1. "*As mentioned above if the filter is pruned, one can simply remove the corresponding BN parameters. Why would one need to do this BN regularization?*"
> > >
> > > Removing BN parameters is perfectly okay, but because of the dependence between parameters, we have to make sure the BN removal does not incur much side-effect. This is why we need the BN regularization (this is similar to that we can do L1-pruning in a one-shot way, but we still prefer to regularize the weights towards zero first so that when we really prune the filters, it does not cause much damage). Also, this is exactly where *our work is different from previous BN-based pruning* (such as network slimming, ICCV, 2017). In network slimming, regularizing BN is to find which filters to prune, while in our case, we already know which filters to prune; regularizing BN is a scheme to mitigate the damage of the pruning.
> > >
> > > "*it is not clear how not regularizing BN can "damage" dynamical isometry. If I understand correctly, if a filter is pruned but the BN parameters are not removed, they have no effect on the network or its Jacobian. What am I missing?*" -- This may not be correct. (1) Consider the BN formulation in Eq. (4): $((W * X - \mu) / \sigma)  \times \gamma + \beta$. If we regularize $W$ to zero without regularizing BN, it will be  $(-\mu / \sigma)  \times \gamma + \beta$. Note this is *not* zero valued. The next layer will still depend on these non-zero values. If we then simply remove these BN parameters (because its associated filters have been pruned), the input for the next layer is *physically changed*. This change can cause error and the error will be accumulated layer by layer, which eventually incurs serious damage to the network isometry and accuracy. (2) In practice, we also observe that, regularizing BN can improve the accuracy before fine-tuning significantly (see our ablation study Tab. 3).
> > >
> > > As suggested, we will add these discussions in our paper to make it more straightforward.
> > >
> > > 2. "*I would recommend adding a brief discussion in the main paper mentioning in practice, there is a range of JSV values that improves performance.*"
> > >
> > > Thanks for the suggestion! We'll add a background section to elaborate on how to use the JSV values properly for practical networks.

---

### Official Review · Reviewer_QYz3 · 2021-11-03

**Correctness:** 3
**Technical Novelty And Significance:** 3
**Empirical Novelty And Significance:** 3
**Recommendation:** 6
**Confidence:** 4

**Main Review:**

Pro:
The authors propose a nice way to maintain dynamic isometry property during pruning: just doing a back-propagation considering the orthogonal constraint. The method is quite intuitive and simple, which I think it’s good.

Cons:
I think the imagenet performance is not quite better than previous methods. The improvement is in general 0.1 points except ResNet50 (3.06x) where the improvement is 0.5 (still not significant). If the authors want to claim their method has larger advantage in larger pruning ratio, there should be more experiments to support such a claim.

The authors argue to use “difference between performance under different learning rate” as a metric to measure whether dynamic isometry is preserved, which in my opinion is not a good measure. Can the author measure the jacobian of each layer?

Questions:

About the claim: “SGD training in finetuning can help recover it”, I’m confused what is the goal of recovery when looking at figure 2. The model seems not to converge to JSV 1, then what information this figure conveys?

What is the L1+KernOrth in Table 1 differs from the proposed method? Is the difference just whether regularize BN parameter? Moreover, there is no explanation of what is the difference between L1 + X v.s. X + L1, which is confusing.

Why this paper only use kernel orthogonality instead of convolutional orthogonality?

Comments:

I think the content would be much easier to read if you can create a 2D plot with the x-axis being the speedup and the y-axis being the accuracy. In its current form, all I can see is the proposed method does not have a clear advantage over the previous method.

Regarding maintaining approximately dynamic isometry, this paper is also related:
[1] deep isometric learning for visual recognition. ICML 2020.

**Summary Of The Paper:**

Dynamic isometry is shown to be a useful property that enable effective gradient propagation through the forward/backward. However, pruning will largely damage such a structure. This paper studies how to maintain the “dynamic isometry” property during pruning. Specifically, after getting an initial assessment of filter importance, the algorithm will maintain the partially kernel orthogonality of the important filters. They also propose to regularize the BN parameters to future boost the performance.

**Summary Of The Review:**

I like the idea of maintaining dynamic isometry during pruning. But there is still great room to improve in both the presentation and analysis.

---

> ### Author Response · Authors · 2021-11-22
> **Responses to  Reviewer QYz3**
>
> We greatly thank Reviewer QYz3 (R1) for your constructive comments! Your concerns are addressed below.
>
> `R1-Q1`: I think the imagenet performance is not quite better than previous methods. The improvement is in general 0.1 points except ResNet50 (3.06x) where the improvement is 0.5 (still not significant). If the authors want to claim their method has larger advantage in larger pruning ratio, there should be more experiments to support such a claim.
>
> `R1-A1`: The advantage of our method looks small (0.1) mainly because (1) the pruning ratio is small, where different methods do not show much difference; (2) we compare to the most recent *state-of-the-art* methods, which are non-trivial to beat. 0.5 top-1 accuracy improvement on ImageNet is actually pretty much in the current filter pruning field. We will add more large-pruning-ratio experiments to support the claim.
>
> `R1-Q2`: The authors argue to use “difference between performance under different learning rate” as a metric to measure whether dynamic isometry is preserved, which in my opinion is not a good measure. Can the author measure the jacobian of each layer?
>
> `R1-A2`: The “difference between performance under different learning rate” is only a *side indicator* to imply that dynamical isometry is well recovered, not the main metric. We use *mean JSV* throughout the paper as the metric to measure whether dynamic isometry is preserved. For example, in Fig. 2, OPP recovers mean JSV from 0.0040 to 3.4875, which can be deemed as recovering dynamical isometry. "Can the author measure the jacobian of each layer?" -- Jacobian of each layer indeed is informative, yet since we want to treat the network as a whole entity, we are more interested in the Jacobian of the *whole network*. In this regard, the mean JSV of the whole network is already provided as metric in the paper.
>
> `R1-Q3`: "“SGD training in finetuning can help recover it”, I’m confused what is the goal of recovery when looking at figure 2. The model seems not to converge to JSV 1, then what information this figure conveys?"
>
> `R1-A3`: JSV 1 is the definition of exact isometry, which is the *ideal case*. In a practical DNN, JSV 1 rarely happens (unless we manually set it to 1). In Fig. 2, we want to show the *similar trend* of OPP (c) to OrthP (b). Particularly, (1) note the initial point of mean JSV of LR 0.001 (it is *larger than 1* in (b) (c); while in (a), it is *close to 0*), (2) note the initial point of test accuracy of LR 0.001, (3) note the mean JSV of LR 0.001 is *above* LR 0.01 for OrthP and OPP, while for (a), the mean JSV of LR 0.001 is *below* LR 0.01. In short, OPP presents a similar trend to OrthP. Thus we claim, OPP is as good as OrthP.
>
> `R1-Q4`: What is the L1+KernOrth in Table 1 differs from the proposed method? Is the difference just whether regularize BN parameter? Moreover, there is no explanation of what is the difference between L1 + X v.s. X + L1, which is confusing.
>
> `R1-A4`: L1+KernOrth means naively combining L1 pruning with Kernel Orthogonalization (Xie et al., 2017): We first conduct the L1-norm pruning and have the slimmer network, which has broken dynamical isometry. Then, during fine-tuning, use the Kernel Orthogonalization (Xie et al., 2017) as a remedy, trying to recover the broken dynamical isometry. As seen, the pruning and DI recovery are *separated*. In our method, pruning is joint with kernel orthogonalization through the proposed regularization target in Fig. 1 (b). Regularizing BN is only a secondary section of our method considering BN is also a part of the network; there is no reason not to regularize it if we are targeting recover DI for the *whole* network.
>
> We explained the difference between L1+X vs. X + L1 in our "Comparison methods" on Page 6: *"There are two possible combination schemes: 1) apply orthogonal regularization methods **before** applying L1-norm pruning, 2) apply orthogonal regularization methods **after** L1-norm pruning, namely, in fine-tuning. "* L1+X means, first applying L1-norm pruning then using X to recover broken dynamical isometry. X+L1 means, first using X to maintain dynamical isometry, then applying L1-norm pruning.
>
> `R1-Q5`: Why this paper only use kernel orthogonality instead of convolutional orthogonality?
>
> `R1-A5`: Although convolutional orthogonality was shown better than kernel orthogonality, it is more complex. This work is the first one that joins structured pruning with orthogonality, so we start with the easier one. Thank R1 for pointing this out! We'll extend our method to convolutional orthogonality in our future version.
>
> `R1-Q6`: I think the content would be much easier to read if you can create a 2D plot with the x-axis being the speedup and the y-axis being the accuracy... This paper is also related: [1] deep isometric learning for visual recognition. ICML 2020.
>
> `R1-A6`: We'll improve the presentation as suggested and cite the related work in ICML 2020.
>
> Hope our feedback can help resolve your concerns!

---

> > ### Comment · Reviewer_QYz3 · 2021-11-30
> > **RE: Responses to Reviewer QYz3**
> >
> > Thanks for the authors' detailed responses. Overall I think it addressed most of my concerns and I'll increase my score to 6.
> >
> > I think this paper is an important first step towards isometry-preserving pruning. But to make a larger impact, there is still significant amount of work to be done especially in the paper organization and experiment results.

---

> > > ### Author Response · Authors · 2021-11-30
> > > **Thanks for Your Feedback!**
> > >
> > > Dear Reviewer QYz3,
> > >
> > > Thanks for kindly improving the score and agreeing with us on that "*this paper is an important first step towards isometry-preserving pruning*"! As suggested, we'll improve the paper organization (e.g., moving the appendix D about how to use mean JSV properly for practical networks to our main paper) and add more experimental results (e.g., the larger pruning-ratio experiments).
> > >
> > > Sincerely,
> > >
> > > Authors

---

### Decision · Program_Chairs · 2022-01-20

**Decision:**

Reject

**Comment:**

The paper proposes a pruning approach that regularizes the gram matrix of convolutional kernels to encourage kernel orthogonality among the important filters meanwhile driving the unimportant weights towards zero. While the reviewers found the proposed method well-motivated and intuitive, they believe that the proposed claims are of limited novelty and are not supported well by the experiments. Analyzing and explaining the effect of different parts of the proposed method, i.e., orthogonalization and regularization of batch normalization parameters, on the accuracy of the pruned models would significantly improve the manuscript.